# OpenReview forum: "Automatic Jailbreaking of Text-to-Image Generative AI Systems for Copyright Infringement"
_ICLR.cc/2025/Conference — Submitted to ICLR 2025_

### Official Review · Reviewer_GDMQ · 2024-10-30

**Soundness:** 3
**Presentation:** 1
**Contribution:** 3
**Rating:** 5
**Confidence:** 4

**Summary:**

This paper shows that commercial Text-to-Image (T2I) systems may be overlooking the risks of copyright infringement, even with basic prompts. While many systems have built-in filters to prevent such violations, the authors’ APGP attack can bypass these protections easily.

The authors use a new approach with a self-generated QA score and a keyword penalty score in its language model optimizer— then no need for any complex weight updates or gradient calculations.

Their tests show that APGP-generated prompts led to copyright issues in 73.3% of cases, even in ChatGPT. Overall, their approach not only makes it easier and cheaper to identify vulnerabilities in T2I models but also helps copyright holders protect their intellectual property more effectively.

**Strengths:**

This paper tackles an interesting topic: the jailbreaking of Text-to-Image generative AI systems for copyright infringement. The overall narrative is both meaningful and engaging. The authors also achieved impressive attack results, even against GPT models.

The authors claim that their approach doesn’t require any weight updates or gradient computations, which I find intriguing. They also designed several loss functions to enhance their attack, and it’s clear that they got solid results with GPT models.

**Weaknesses:**

The overall story and topic are definitely interesting, especially given the impressive results the authors achieved. However, the paper itself isn’t well-written; there are too many typos and unclear statements that need to be addressed. For example,

1. Figure 1a doesn’t seem necessary.
2. Figure 2 is hard to interpret; it’s unclear why weight updates or gradient updates aren’t needed, and how to update the instructions isn’t explained well.  maybe it's an overclaim because you still need to update the instructions.
3. There are too many symbols in the equations that aren’t clearly defined. For instance:
   - What does "m" refer to in line 210?
   - In line 210, "v" is used to represent LLM, but then in line 231, "v" represents the encoder.
   - What’s the expression for \( S_k \)?
   - "f2" isn’t defined in line 301.
   - There are reference errors for Figure B.2 in lines 344 and 351, and the image formatting at the bottom of page seven is clearly off.
   - Additionally, there are reference errors for Table 9 on line 407.

On top of that, the ablation study in Figure 9 is incomplete. What happens if you remove the two score functions?

**Questions:**

For the unlearning experiments, it seems that the attack is not directly targeting GPT, but rather the diffusion model with unlearning, right?  This may not be very convincing; maybe the unlearning algorithm is just bad on this model.

---

> ### Author Response · Authors · 2024-11-23
> **Revised manuscripts / Update figure / Ablation experiment**
>
> Thank you for your time and constructive feedback. We appreciate your comments and have made every effort to address them in the revision. If you have any further concerns, please feel free to reach out for discussion.
>
> **[W1. Editorial comments-1. Figure 1a]** Figure 1a doesn’t seem necessary.
> - We removed Figure 1a and updated Figure 1 and 2.
>
> **[W2. overclaim updates instruction]** Figure 2 is hard to interpret; it’s unclear why weight updates or gradient updates aren’t needed, and how to update the instructions isn’t explained well. maybe it's an overclaim because you still need to update the instructions.
> - **We believe this may be a misunderstanding of our work.** Our approach **does not require any weight or gradient updates**, as we **adjust the instructions or descriptions** solely **based on language model inference** using the given prompt as [1,2] (Lines 773-897).
> - Specifically, we provide the language model with a previous output and its corresponding score, then **ask the LLM to revise the output (instruction/description) to achieve a higher score**. We refer to this process as an 'update.' The detailed pipeline can be found in Figure 14.
>
> [1] Large Language Models as Optimizers \
> [2] Improving Text-to-Image Consistency via Automatic Prompt Optimization
>
> **[W3. Editorial comments-2. Formula]** There are too many symbols in the equations that aren’t clearly defined.
> - **We reflect all your comments.**
>   * What does "m" refer to in line 210?
>       - &#8594; It is an index of question and answer pairs (line 200).
>   * In line 210, "v" is used to represent LLM, but then in line 231, "v" represents the encoder.
>       - &#8594; Sorry for misuse of the term. We revise the vision encoder to “e_v”.
>   * What’s the expression for ( S_k )?
>       - &#8594; It is a keyword penalty in the score function (Line 190).
>   * "f2" isn’t defined in line 301.
>       - &#8594; It is a LLM model that refines the description (Line 186)
>   * There are reference errors for Figure B.2 in lines 344 and 351, and the image formatting at the bottom of page seven is clearly off.
>       - &#8594; We revised the reference error to Section B.2 Figure 17.
>   * Additionally, there are reference errors for Figure 9 on line 407.
>        - &#8594; We revise the reference error to Table 7.
>
> **[W4. Ablation study in Table 7]** The ablation study in Table 7 is incomplete.
> - We **conducted additional ablation experiments** in the following Table.
> - However, as shown in the following table, **the block rate trend is consistent with Table 7**, where keyword penalties contribute the most to the block rate.
> - Additionally, **leaving the keyword penalty while removing the QA score allows prompts to bypass** block censorship more easily; however, this results in the generation of **general outputs that do not infringe on copyright, as illustrated in Figure 5**.
> - As each score component is independent, we believe Table 7 demonstrates enough ablation study to see which component plays important roles in the block rate.
>
> ||Character|Art|Average|
> |-|-|-|-|
> |S|26.67|24.00|30.28|
> |wo S_k, S_qa|51.43|20.00|35.71|
> |wo S_k, S_i|62.86|28.00|45.43|
> |wo S_qa, S_i|37.14|20.0|28.57|
>
> **[Q1. unlearning experiment]** For the unlearning experiments, it seems that the attack is not directly targeting GPT, but rather the diffusion model with unlearning, right? This may not be very convincing; maybe the unlearning algorithm is just bad on this model.
> - **No, this is a misunderstanding of our experimental setting.** We employ an open-sourced unlearning **checkpoint of the diffusion model** that is trained by authors and tested our APGP generated prompt on that diffusion model. This experiment demonstrates not only **ineffectiveness of unlearning** but also shown the **transferability of our high-risk prompt to diffusion models**.

---

> > ### Comment · Reviewer_GDMQ · 2024-11-27
> > **reply**
> >
> > Thanks for the reply!
> >
> > >  ask the LLM to revise the output (instruction/description) to achieve a higher score. We refer to this process as an 'update.'
> >
> > so there is no optimization? how do you know if the new instruction is better than the previous one?
> > Do you just pick the one with good scores? Then I wouldn't call it 'update', maybe just 'generate a new description'.
> >
> > > As each score component is independent, we believe Table 7 demonstrates enough ablation study to see which component plays important roles in the block rate.
> >
> > I wonder if combining two or three really makes a big difference. So a detailed ablation study is always appreciated.
> >
> > > We employ an open-sourced unlearning checkpoint of the diffusion model that is trained by authors and tested our APGP generated prompt on that diffusion model
> >
> > I am less interested in, or convinced by, these experiments on unlearning. It could be that unlearning is just bad, making it easy to attack; or it could be that unlearning is perfect, yet open to attack. We don't know which is true.

---

> > > ### Author Response · Authors · 2024-11-28
> > > **Clarification on the term 'update' / Explanation of the unlearning approach**
> > >
> > > > so there is no optimization? how do you know if the new instruction is better than the previous one? Do you just pick the one with good scores? Then I wouldn't call it 'update', maybe just 'generate a new description'.
> > > * We believe there may be a misunderstanding regarding our approach.
> > >
> > > * **Definition of 'optimization' and ‘update':** Our method does **not generate a single description in one step**. Instead, we iteratively refine the description across multiple steps so that each subsequent iteration achieves a higher score than the previous one. This iterative process aligns with the concept of optimization and gradual improvement until the final result is reached.
> > > * The **LLM acts as an optimizer during this process**, working to maximize the score with each step. The term 'update' reflects this iterative refinement process, as described in [1,2]. It is not merely about selecting the description with the highest score but involves step-by-step optimization, **where the descriptions evolve to achieve better performance**.
> > > * **Clarification on optimization:** While this process **may differ from traditional weight optimization, it still constitutes optimization in the context of description refinement**. The final description used is the result of this optimized iterative process, which is why we refer to it as an 'update.'
> > >
> > > [1] Yang et al., Large Language Models as Optimizers, ICLR 2024 \
> > > [2] Manas et al., Improving Text-to-Image Consistency via Automatic Prompt Optimization, TMLR 2024
> > >
> > > > I wonder if combining two or three really makes a big difference. So a detailed ablation study is always appreciated.
> > > * Thank you for recognizing our efforts. We will include the experimental results in the manuscript, as detailed results in our previous response.
> > >
> > > > I am less interested in, or convinced by, these experiments on unlearning. It could be that unlearning is just bad, making it easy to attack; or it could be that unlearning is perfect, yet open to attack. We don't know which is true.
> > > * We believe unlearning is good work, yet open to attack. As the unlearning paper [1] was published at ICCV 2023 and the authors conducted extensive evaluations across diverse datasets and settings, we do not claim that unlearning is just bad. Instead, we propose that there is still room for improvement in achieving full unlearning of the targeted concept.
> > > * In this experiment, **our goal is to show that our prompt is effective and remain robust against several straightforward defense strategies, such as filtering, prompt rephrasing, or unlearning.**
> > >
> > > [1] Kumari et al., Ablating concepts in text-to-image diffusion models., ICCV 2023
> > >
> > > If you have any further questions, please feel free to reach out. If our responses have addressed your main concerns, we kindly hope you will consider reflecting this in the final scores.

---

> ### Author Response · Authors · 2024-11-25
> **Gentle reminder / [Revised manuscripts / Update figure / Ablation experiment]**
>
> **This is a gentle reminder regarding your initial concerns.**
>
> We have addressed them through additional explanations and experiments, and we kindly invite you to review our response. If you have any remaining concerns or suggestions, **we would be delighted to engage in a final discussion to ensure a fair review process.**
>
> * The manuscript has been revised and thoroughly proofread.
> * All editorial comments have been addressed, and the figures have been revised for better clarity and understanding.
> * An additional score function ablation experiment is provided.
>
> We greatly appreciate your time and effort and look forward to hearing from you.

---

### Official Review · Reviewer_Lo6a · 2024-11-03

**Soundness:** 3
**Presentation:** 2
**Contribution:** 2
**Rating:** 6
**Confidence:** 3

**Summary:**

The authors focus on assessing and challenging the copyright infringement safeguards in commercial T2I systems such as ChatGPT, Copilot, and Gemini. They create a dataset, termed VioT, comprising images of copyrighted content (characters, logos, products, and artworks) and devise an Automated Prompt Generation Pipeline (APGP) that uses language models to generate prompts that circumvent copyright safeguards. The study finds that, despite existing safety measures, models are vulnerable to producing unauthorized reproductions, demonstrating that only ChatGPT consistently blocks such prompts at a rate of 96.25%. The APGP method reduces ChatGPT’s block rate to 6.25%, highlighting the need for more robust protection mechanisms.

The authors suggest that current defenses, like post-generation filtering and machine unlearning, are inadequate, indicating a critical need for improved defense strategies in T2I models.

**Strengths:**

1. Copy infringement is an important problem.
2. VioT dataset: Providing a dataset that the future methods can compare with is useful. 20 images in each of the 4 catergoreis were provided.
3. Human Evaluation gives the approach credibility. Infact introducing metric for evaluation is also useful.

**Weaknesses:**

1. The presentation in the experiment section is not upto par with ICLR. The figures and text should be arranged properly.
2. The idea is similar to treating VLM and LLM as two agents helping to jailbreak the T2I diffusion model. How is approach different from [1].
3. 1. VioT dataset: 20 images in each of the 4 catergoreis were provided. However I feel the number of images is small to text the validity of the approach.
4. Lack of scoring function ablation details to understand each of its contribution. Why is there a linear addition? Is there no normalization of the values? Such details are very important to understand the scoring function.

Further details related to weakness are asked using questions below.



1. Dong, Yingkai, et al. "Jailbreaking Text-to-Image Models with LLM-Based Agents." arXiv preprint arXiv:2408.00523 (2024).

**Questions:**

I really like the idea of using VLM and LLM as two agents to jailbreak T2I. However I have a few questions:
1. I see that the LLM observes the instructions and score to optimize future instructions. Why is a single scalar score given to LLM. Why is scoring not given separately and the LLM is asked to improve each of the scores. Such analysis and baseline of the scoring function is important. Is there any ablation done for contribution of individual parts of the scoring function and how was the final contribution of each scoring function finally decided?
2. Why is not a single VLM enough for our approach? What I mean is, what if we provide the score and the system prompt (to make an VLM act like an optimizer) and ask the VLM itself to generate descriptions such that improves the score ? Why is reason for not trying that and introducing a LLM separately ?
3. Why the gradient based methods to optimizing prompts for jailbreaking LLM have not been tried for T2I models? (It could have helped in developing a better agentic framework)

---

> ### Author Response · Authors · 2024-11-23
> **Revised manuscript / Comparisons with [1]  / Increased dataset size / Score function ablation**
>
> Thank you for your time and constructive feedback. We appreciate your insightful comments and have done our best to address the initial concerns. If you have any further questions or suggestions, please feel free to reach out for discussion.
>
> **[W1. Formatting]** The figures and text should be arranged properly.
> - We apologize for the inconvenience. We've conducted another round of proofreading to improve the writing quality across the entire manuscript. As **ICLR allows updating the pdf, we've updated the final PDF for better clarity**. Please review the updated version.
>
> **[W2. [1] difference]** The idea is similar to treating VLM and LLM as two agents helping to jailbreak the T2I diffusion model. How is the approach different from [1].
> - While the high-level concept of using LLMs or VLMs as agents is similar, the **jailbreaking task, target models and method differ significantly**.
> - [1] focuses on generating harmful images in T2I models with a single-layer safety filter, such as Stable Diffusion or DALLE. In contrast, our approach targets generating copyrighted content in T2I services with multi-layered filtering systems, like ChatGPT or Copilot.
> - Additionally, **the methods differ**: [1] uses chain-of-thought reasoning and in-context learning to iteratively learn from previous attempts on the target model, whereas our approach relies on a single attempt guided by predefined score functions.
> - We cited **this concurrent work** in the related work section.
>
> **[W3. Lack of dataset]** However I feel the number of images is small to text the validity of the approach.
> - We conducted an additional 70 images, bringing the total to 140 images. As shown in the following table, the **block rate remains significantly low even with an enlarged dataset**, demonstrating consistent risks in T2I systems.
> - We believe that 140 images are sufficient to highlight the risks of current T2I systems. However, if you still have concerns about the dataset size and its ability to demonstrate our effectiveness, we are willing to collect additional data. Please let us know the sufficient size.
> - **Additional data can be found in the supplementary folder.** The images are listed at the bottom.
> ||Product|Logo|Character|Art|Average|
> |-|-|-|-|-|-|
> |70 images|2.67$\pm$2.89|25.00$\pm$8.66|38.34$\pm$25.16|33.34$\pm$12.58|24.84$\pm$12.32|
> |140 images|5.71$\pm$2.86|15.24$\pm$5.95|26.67$\pm$22.19|24.00$\pm$12.00|17.90$\pm$10.75|
>
> **[W4. scoring function ablation]** Lack of scoring function ablation details to understand each of its contributions. Why is there a linear addition? Is there no normalization of the values?
> - It is linearly added because we believe **each component is independently important** to the final score. There is no normalization of the values.
> - In the scoring function ablation, we simply remove each score term in Table 7.
> - To demonstrate the contribution of each score function, we conducted a score ablation in Table 7. As seen in Table 7, the keyword penalty (S_k) plays the most significant role in bypassing the safeguard, while the QA score term contributes to generating precise images, as illustrated in Figure 5.

---

> ### Author Response · Authors · 2024-11-23
> **Question: single scalar / single VLM / gradient based methods**
>
> **[Q1. Single scalar in score function]**  Why is a single scalar score given to LLM. Why is scoring not given separately and the LLM is asked to improve each of the scores.
> - We selected **the single scalar score given to the LLM, following the approach of previous work [1,2]** in the instruction-following task.
> - However, your comments were very insightful. As a result, we conducted an additional experiment using an independent score set rather than a single scalar. **This approach is also optimized well for finding high-risk prompts**, as shown in the table below. However, **we think we need additional reasoning ability to fully leverage each independent score terms**.
>
> ||Product|Logo|Character|Art|Average|
> |-|-|-|-|-|-|
> |Single scalar S|5.72|15.24 |26.67 |24.00 |17.90|
> |Independent score term|5.72|8.58|25.72|28.00|17.01|
>
> [1] Large Language Models as Optimizers \
> [2] Improving Text-to-Image Consistency via Automatic Prompt Optimization
>
> **[Q2. Why isn't a single VLM not enough?]** Why is not a single VLM enough for our approach? What I mean is, what if we provide the score and the system prompt (to make an VLM act like an optimizer) and ask the VLM itself to generate descriptions such that improves the score ? Why is reason for not trying that and introducing a LLM separately ?
> - Thanks for your insightful comments. When we use VLM, it could make a more powerful jailbreaking approach. However, **we might need a slightly different score function and prompt template for the VLM since we give an image as a condition**.
> - It is possible to replace the LLM with a VLM in step 2, but **we believe strong language performance is more relevant to update the prompts adequately** based on the given score. Furthermore, using a VLM is computationally **more expensive**, as it requires processing images as an additional input.
>
> **[Q3. Why the gradient based methods to optimizing prompts for jailbreaking LLM have not been tried for T2I models?]** Why the gradient based methods to optimizing prompts for jailbreaking LLM have not been tried for T2I models? (It could have helped in developing a better agentic framework)
> - First, since we are targeting black box T2I systems, ChatGPT or Copilot, not single T2I models like stable diffusion or Dalle, **which do not provide a whole system as a single API which makes it difficult to collect their feedbacks during the optimizing**.
> - Furthermore, we found gradient-based methods to be computationally inefficient compared to our approach, which **needs training and weight updates** of all models.
> - Additionally, many gradient-based jailbreaks [1,2] tend to produce unnatural prompts that can be easily filtered inside the system.
>
> [1] Universal and transferable adversarial attacks on aligned language models \
> [2] Autodan: Automatic and interpretable adversarial attacks on large language models.

---

> > ### Author Response · Authors · 2024-11-25
> > **Gentle reminder / [Revised manuscript / Comparisons with [1] / Increased dataset size / Score function ablation]**
> >
> > **This is a gentle reminder regarding your initial concerns.**
> >
> > We have addressed them through additional experiments and revisions, and we kindly invite you to review our response.
> > If you have any remaining concerns or suggestions, **we would be delighted to engage in a final discussion to ensure a fair review process.**
> >
> > **[Summary of responses]**
> > * The manuscript has been revised and thoroughly proofread.
> > * We conducted a comparison with [1], noting that [1] addresses a different task, method, and target as a jailbreaking attack. Additionally, [1] is concurrent work with ours.
> > * The dataset size has been increased, yielding consistent results.
> > * Detailed information about the score function ablation experiment has been provided.
> >
> > We greatly appreciate your time and effort and look forward to hearing from you.

---

> > > ### Author Response · Authors · 2024-11-28
> > > **Discussion phase has been extended**
> > >
> > > As the discussion phase has been extended, we would like to gently remind you of your initial concerns.
> > >
> > > We have addressed them through additional experiments and revisions, and we kindly invite you to review our responses. If you have any remaining questions or suggestions, we would be happy to engage in the discussion to ensure a thorough and fair review process.
> > >
> > > We sincerely appreciate your time and effort and look forward to your feedback.

---

> > > > ### Comment · Reviewer_Lo6a · 2024-12-03
> > > >
> > > > I don't have any other question. Thank you for your answers.  I have raised my score.

---

### Official Review · Reviewer_znT6 · 2024-11-03

**Soundness:** 3
**Presentation:** 2
**Contribution:** 3
**Rating:** 6
**Confidence:** 2

**Summary:**

This paper evaluated the safety of the commercial T2I generation systems on copyright infringement with naive prompts. The paper also proposed a stronger automated jailbreaking pipeline for T2I generation systems, which produced prompts that bypass their safety guards.

**Strengths:**

1. Comprehensive evaluation results
2. The paper also tested some simple defenses to mitigate their attack
3. The automated prompt generation process to stress-test the VLM for copyright issue is an important research question.

**Weaknesses:**

1. L50, "To the best of our knowledge, there is no work on quantitative evaluation of the copyright violation by the commercial T2I systems". Can you talk about the relationship between your work and the Glaze tool [1]? The Glaze tool also aims to protect the copyrighted and private images created.
2. Suggest to recreate Fig. 1 and consider combining Fig. 2 with Fig. 1. Fig. 2 seems to be the major selling point of the paper, while I cannot clearly tell from Fig. 1, which generated image is copyrighted versus safe to output to users.
3. L193, a(n) automated
4. Why is this tool copyright specific? I feel like the prompt generation pipeline is agnostic to the type of the attack? Maybe I missed something here.
5. What is a potential solution for/defense against such type of prompt attack?

[1] https://glaze.cs.uchicago.edu/what-is-glaze.html

**Questions:**

See weakness

---

> ### Author Response · Authors · 2024-11-23
> **Relation with Glaze / Figure update / Specific tool / Potential defense**
>
> Thank you for your time and constructive feedback. We greatly appreciate your insightful comments and questions. We have done our best to address them, and if you have any further concerns, please feel free to reach out for further discussion.
>
> **[W1. Relation between Glaze and Ours]** Can you talk about the relationship between your work and the Glaze tool [1]? The Glaze tool also aims to protect the copyrighted and private images created.
> - The Glaze tool aims to protect copyrighted content **during the training data collection phase** (Line 135), preventing its features by adding adversarial noises into the content from being learned by T2I models.
> - However, current T2I systems did not use Glaze when training the model. Instead, they employed alignment learning to refuse responses to copyright infringement requests.
> - Our work demonstrates that **current alignment does not fully protect against copyright infringement**.
>
> **[W2. Editorial comment-Figure]** Suggest to recreate Fig. 1 and consider combining Fig. 2 with Fig. 1.
> - We revised the figure 1 and 2 in the pdf.
>
> **[W3. specific tool]** Why is this tool copyright specific? I feel like the prompt generation pipeline is agnostic to the type of the attack?
> - Our approach is **not limited to a specific type of attack but can be applied to various scenarios that involve safeguarding specific content** (like copyrighted content, or celebrities’ image).
> - The APGP technique leverages the characteristics of T2I models to revoke specific images without relying on directly linked keywords. This allows it to bypass safeguards, which are typically aligned based on the keyword, thereby revealing the original content.
> - As the reviewer pointed out, **this tool can also be adapted to other types of attacks, such as those involving publicity rights leakage**, as shown in Figure 11.
> - As a result, while the prompt generation pipeline itself is agnostic to the type of attack, **copyright protection serves as a practical demonstration of the tool's effectiveness.**
>
> **[W4. Potential defense/solution]** What is a potential solution for/defense against such type of prompt attack?
> - Glaze could indeed serve as an effective initial protection for copyrighted content.
> - Additionally, implementing extra layers of filtering systems or utilizing our high-risk APGP prompts for alignment learning could further enhance safeguard measures.
> - If you find **this potential defense and solution reasonable, we can include these points in the Ethical Statement section** as well.

---

> > ### Author Response · Authors · 2024-11-25
> > **Gentle reminder / [Relation with Glaze / Figure update / APGP tools / Potential defense]**
> >
> > **This is a gentle reminder regarding your initial concerns.**
> >
> > We have addressed them through additional explanations, and we kindly invite you to review our response. If you have any remaining concerns or suggestions, **we would be delighted to engage in a final discussion to ensure a fair review process.**
> >
> > * We provide relation between Glaze and ours.
> > * The figures have been revised for better clarity and understanding.
> >
> > We greatly appreciate your time and effort and look forward to hearing from you.

---

> > > ### Comment · Reviewer_znT6 · 2024-11-27
> > >
> > > Thank you for your response! I apologize for the late reply. All my concerns are addressed and I will keep my positive rating.

---

> > > > ### Author Response · Authors · 2024-11-28
> > > >
> > > > Thank you for your acknowledgment and positive evaluation. We are delighted to know that our revisions have successfully addressed your initial concerns.

---

### Official Review · Reviewer_nAdo · 2024-11-03

**Soundness:** 3
**Presentation:** 3
**Contribution:** 3
**Rating:** 6
**Confidence:** 4

**Summary:**

The authors perform a study about the tendency of SOTA T2I models to produce copyrighted content. They do so by:
- building a copyright violation dataset for T2I models, with characters, logos, products and arts.
- producing naive prompts and a jailbreaking procedure
- analysing the successfulness of both attacks and defences in the generation of copyrighted content across SOTA models. Concluding defences are currently inadequate.
- the automated jailbreaking procedure uses an ageintic approach which is interesting.

**Strengths:**

- Originality: the work is not extremely novel since several papers exist that use LLMs to jailbreak or induce regurgitation of training data of other models (e.g. [1,2]), one of which does it for memorization uncovering. Similarly the optimization procedure that is proposed is not extremely novel. However I do not know other works that use MLLMs for this specific purpose.
-  Clarity: The paper writing is clear
- Quality: the methodology is good and the experiment quality is sufficient.
- Significance: The problem is of obvious relevance to companies. The introduced dataset and jailbreaking procedure have the potential of being useful.

[1] https://arxiv.org/html/2312.02119v3

 [2] https://arxiv.org/abs/2403.04801

**Weaknesses:**

- The authors counterargue the idea of using a coopyright detection model, suggesting that since there's no open sourced one then it's not practical. Would it be possible to just use MLLMs themselves as filters of copyrighted contents? An LLM could probably guess a good list of entities that are copyrighted (or find them in a catalogue) given the prompt and then an MLLM can simply verify the presence or absence of the copyrighted content in the image.

**Questions:**

Minor observation, The figure after Figure 4 is obviously misplaced.

---

> ### Author Response · Authors · 2024-11-23
>
> Thank you for your time and constructive feedback. We appreciate your acknowledgment of the originality, clarity, experimental quality, and significance of our research. If you have any further concerns or suggestions, please feel free to reach out, and we would be happy to discuss them further.
>
> **[MLLM detection]** The authors counter argue the idea of using a copyright detection model, suggesting that since there's no open sourced one then it's not practical. Would it be possible to just use MLLMs themselves as filters of copyrighted contents? An LLM could probably guess a good list of entities that are copyrighted (or find them in a catalogue) given the prompt and then an MLLM can simply verify the presence or absence of the copyrighted content in the image.
> - Thank you for the suggestion. While using MLLMs to filter copyrighted content is an interesting idea, copyright detection remains complex. MLLMs might identify some copyrighted elements, but they can still **miss nuanced cases or produce false positives**. Therefore, applying MLLMs to copyright detection requires additional prompting or reasoning approaches to create a reliable model.
> - The lack of **reliable, open-source copyright detection models** underscores the limitations of **simply applying previous harmful image T2I jailbreaking approaches** to our copyright jailbreaking problem.
>
> **Minor observation, The figure after Figure 4 is obviously misplaced.**
> - We revised the manuscripts. Thanks for the comment.

---

> > ### Author Response · Authors · 2024-11-25
> >
> > **This is a gentle reminder regarding your initial concerns.**
> >
> > We have addressed them through additional explanations, and we kindly invite you to review our response.
> > If you have any remaining concerns or suggestions, **we would be delighted to engage in a final discussion to ensure a fair review process.**
> >
> > We greatly appreciate your time and effort and look forward to hearing from you.

---

> > > ### Comment · Reviewer_nAdo · 2024-11-27
> > > **Response to authors**
> > >
> > > I acknowledge I have read the rebuttal (thanks for addressing my concerns) and other reviewer's comments and questions. For the moment I maintain the current positive assessment of the work.

---

> > > > ### Author Response · Authors · 2024-11-28
> > > >
> > > > Thank you for your response. We are delighted to see that our revisions have addressed your initial concerns.

---

### Official Review · Reviewer_TRPN · 2024-11-04

**Soundness:** 3
**Presentation:** 3
**Contribution:** 2
**Rating:** 6
**Confidence:** 4

**Summary:**

The paper primarily studies the critical issue of copyright infringement in text-to-image (T2I) models. Initial analysis showed that popular T2I systems such as Midjourney, Copilot, and Gemini are highly vulnerable to copyright violations even when using simple jailbreak prompts. While ChatGPT had a block rate of around 84% on simple prompts, the authors crafted an Automated Prompt Generation Pipeline (APGP) which significantly reduced the ChatGPT’s block rate to around 6%. To summarize the contributions: (1) the authors highlighted a serious safety issue: showing that state-of-the-art T2I models can be easily jailbroken using optimized prompts without needing access to model gradients, and 2) created an annotated dataset, VioT, for evaluating copyright violations in T2I models.

**Strengths:**

1. The finding that these widely used T2I models can be jail-broken using a simple prompt optimization approach is valuable to the community for further research on designing defense mechanisms.

2. The overall paper presentation is quite good, with the research problem well-articulated to the reader (Except Figure 5 on page 7 which has some small formatting issues). Further, each component of the proposed jailbreak attack has also been well-motivated.

3. Evaluation on the proposed VioT dataset clearly shows the efficacy of the proposed jailbreak framework. It is nice to see evaluations using both human and automatic metrics, which strengthens the experimental evaluations.

**Weaknesses:**

1. There are two stages of optimization in APGP, first using the VLM to search for the seed prompt and then again revising the prompt based on some defined scores. The authors should provide a comparison of the latency of their approach against other jailbreak methods. It is essential to understand the computational overhead of the approach for practical applications.

2. Based on, Figure 13 (Appendix A.1), the VioT dataset has only 70 images. I think the evaluation of the proposed framework on only 70 images is not very convincing to demonstrate its effectiveness. I request the authors to clarify if I am missing something.

3. [Minor] Although the paper has ablations on each component, it would further strengthen the draft if the authors could include an ablation on the LLM used for optimizing the prompt, which has been currently set as GPT-3.5-Turbo.

**Questions:**

To summarize the weakness, my major concerns are regarding the overall latency for generating the jailbreak prompts (see Weakness 1) and the limited size of the evaluation dataset (see Weakness 2).

---

> ### Author Response · Authors · 2024-11-23
> **Latency comparisons / Increased dataset size**
>
> Thank you for your time and constructive feedback. We appreciate your acknowledgment of our interesting initial observations, well-motivated approach, clear presentation and strong experimental evaluation. If you have any further concerns, let us know to discuss.
>
> **[W1. Latency of APGP]** The authors should provide a comparison of the latency of their approach against other jailbreak methods
> - The average latency for each image using **APGP is 3.71 minutes**. In comparison, the average latency for the most popular language jailbreaking using **PAIR [1] and GCG [2] is 0.5 minutes and 108 minutes** per prompt, respectively. This demonstrates that our approach is reasonable for text-to-image models.
> - Additionally, please note that there are **no other known jailbreak methods for copyright contents**.
> - For specific latency, step 1 takes 18.29 seconds and step 2 takes 33.49 seconds, with 2.89 seconds for updating prompts and 6.12 seconds for scoring all previous top five prompts. Overall, three iterations of step 1 and five of step 2 result in a total runtime of 3.71 minutes. Using fast inference methods can significantly reduce this latency even more.
>
> [1] Jailbreaking Black Box Large Language Models in Twenty Queries \
> [2] Universal and transferable adversarial attacks on aligned language models
>
> **[W2. Small dataset]** Based on Figure 13 (Appendix A.1), the VioT dataset has only 70 images. I think the evaluation of the proposed framework on only 70 images is not very convincing to demonstrate its effectiveness.
> - We conducted an additional 70 images, bringing the total to 140 images. As shown in the following table, the **block rate remains significantly low even with an enlarged dataset**, demonstrating consistent risks in T2I systems.
> - We believe that 140 images are sufficient to highlight the risks of current T2I systems. However, if you still have concerns about the dataset size and its ability to demonstrate our effectiveness, **we are willing to collect additional data. Please let us know the sufficient size.**
> - **Additional data can be found in the supplementary folder**. The images are listed at the bottom.
>
> ||Product|Logo|Character|Art|Average|
> |-|-|-|-|-|-|
> |70 images|2.67$\pm$2.89|25.00$\pm$8.66|38.34$\pm$25.16|33.34$\pm$12.58|24.84$\pm$12.32|
> |140 images|5.71$\pm$2.86|15.24$\pm$5.95|26.67$\pm$22.19|24.00$\pm$12.00|17.90$\pm$10.75|
>
> **[W3. Other LLM]** [Minor] Although the paper has ablations on each component, it would further strengthen the draft if the authors could include an ablation on the LLM used for optimizing the prompt, which has been currently set as GPT-3.5-Turbo.
> - We additionally employ GPT-4o-mini and the prompt risks even increased as shown in the Table.
> - Since our approach is **a black-box attack that does not rely on any weights or feedback from the target model**, it remains applicable as long as the **LLM demonstrates sufficient language performance**.
>
> ||Product|Logo|Character|Art|Average|
> |-|-|-|-|-|-|
> |GPT3.5-turbo|5.72 |15.24 |26.67 |24.00 |17.90|
> |GPT-4o-mini|2.86|14.29|11.43|20.00|12.14|

---

> > ### Comment · Reviewer_TRPN · 2024-11-24
> > **Thanks for the additional results**
> >
> > Thank you for providing a comparison of the latency of APGP with other approaches. The latency for APGP appears reasonable.
> >
> > The evaluation on a larger dataset size provides stronger evidence of the approach's efficiency. The observed improvement in the average block rate over 140 images is a good sign.
> >
> > I also appreciate the additional ablations conducted on other GPT models.
> >
> > The rebuttal has covered all my concerns, hence I would like to keep my score and recommend accept.

---

> > > ### Author Response · Authors · 2024-11-25
> > >
> > > Thank you for your response. We are delighted to see that our revisions have addressed your initial concerns.
> > >
> > > We kindly request your acknowledgment of the resolved concerns regarding the data size during the reviewer-AC discussion, as these were also raised by Reviewer Z4wV and Reviewer Lo6a.
> > >
> > > Please let us know if there are any further aspects you would like to discuss.

---

### Official Review · Reviewer_Z4wV · 2024-11-04

**Soundness:** 2
**Presentation:** 1
**Contribution:** 2
**Rating:** 5
**Confidence:** 4

**Summary:**

The authors study jailbreaking commercial text-to-image systems to produce copyright infringing outputs, and highlight the vulnerability of these systems. The authors benchmark 4 systems on a dataset of copyright images they labeled (70 images). The authors show that current t2i systems produce copyright infringing outputs readily, with the except of ChatGPT. The authors propose an automatic jailbreak pipeline to generate copyright infringing images, by prompting LLM to literately refine a jailbreak prompt. The author follow a setup close to OPRO (Yang et al.), using LLM as an optimizer and uses CLIP score as optimization feedback. Unlike prior work, a classifier is not required for the jailbreak method but just a target image. The jailbreak pipeline improves over naive prompting and prior work by reducing the block rate of copyright generation, and achieving higher copyright infringement evaluation based on the human study.

**Strengths:**

1. The authors propose a practical pipeline and demonstrates its efficacy. The ablation experiments show how the different components in the score contribute to the performance.
2. The authors point out the lack of robustness against copyright generation of t2i systems, with additional experiments showing the same vulnerability in one concept erasure method.
3. The authors discuss the societal implication of their work and motivates the topic of study well.

**Weaknesses:**

1.  The authors acknowledge that 'our approach has the limitation that the violation rate does not always reproduce the same due to the randomness of the commercial T2I systems' (line 485). Given the small dataset size (70 images), how many iterations were run for the experiments in the tables? Could the authors report block rates and evaluation scores based on averages across multiple iterations?
2. The paper has a high number of formatting, stylistic, and wrong element reference issues. The paper would benefit greatly from another round of proofreading for writing quality and clarity. The list below is not exhaustive.

typo/wrong element reference:
- Line 407: should be linking Table 5 instead of Table 9
- Figure 5 is not present in the paper, even though it was referenced in main text
- "Charcater" in Table 7
- Line 661: "There are 20 images in each category, as shown in Table 13." The art category has 10 samples, not 20.

style/format:
- Many sentences are awkwardly constructed and/or have grammatical errors. For example: "Furthermore, not only generating the
contents, the contents are exceptionally similar to the original IP content as shown in Figure 3"  (line 354). "This work has been
deemed exemption by IRB (IRB-2x-3xx) in anonymous" (line 903). "We show that the majority of commercial T2I systems result in copyright violation" (line 127). "Gemini-pro blocks all human-included generation in the current version which may block content not due to its harmfulness" (line 221).
- Image was cut off on page 7
- Large gap of spacing in the middle of page 8
3. The authors have conducted a human study to understand whether study participants consider generated images to be have copyright infringement issues. However, there does not seem to be discussion around how study participants are trained towards differentiating copyright infringement and fair use. The human study provides more insight around participant perception of copyright infringement, than actual determination of violation determination.
4. Commercials T2I models are updated continuously. Please include the release date or version of each T2I model.
5. One of the paper's contribution is dataset, though the dataset was not included as an anonymized link for review.

**Questions:**

Q1: "Identical violations" was mentioned twice in the paper but not defined. In lines 359-360, authors state that "Upon examining the images classified as identical violations, it was found that over 50% were deemed to be cases of copyright infringement in product and logo." If identical violations refer to generating nearly identical images, the number of over 50% being deemed as infringement seems low.

Q2:  Lines 371-372: "In Figure 4, 42.19% of the generated images correctly answer more than seven questions." What are the seven questions?

Q3: How many iterations of experiments are run for each copyright content in the dataset, for each table? Since these systems have randomness, how is the variance accounted for?

---

> ### Author Response · Authors · 2024-11-23
> **Multiple runs / Increased dataset size / Proofread / Dataset released**
>
> Thank you for your valuable feedback and constructive comments. We have carefully addressed all your concerns through additional proofreading and supplementary experiments. Please let us know if you have any further questions or suggestions; we are happy to discuss them.
>
> **[W1. Q3. multiple iterations / small dataset]** Could the authors report block rates and evaluation scores based on averages across multiple iterations?
> - We conducted **three trials and averaged the block rates** shown in the table below. The results indicate consistent vulnerabilities in the T2I systems across multiple runs.
> - Additionally, we experimented with 140 images to verify our approach, as shown in the following table. Additional data can be found in the supplementary folder. The images are listed at the bottom.
>
> |140 images|Product|Logo|Character|Art|Average|
> |-|-|-|-|-|-|
> |Copilot|6.67$\pm$6.60|7.62$\pm$5.95|1.90$\pm$3.30|29.33$\pm$8.33|**11.38**$\pm$6.04|
> |ChatGPT|5.71$\pm$2.86|15.24$\pm$5.95|26.67$\pm$22.19|24.00$\pm$12.00|**17.90**$\pm$10.75|
>
> - Despite the variance, **our approach still has a significant success rate in bypassing safeguards**, and T2I systems continue to exhibit copyright infringement using our prompts. We believe these results validate the effectiveness of our work and highlight the importance of addressing this real-world problem.
>
>
> **[W2. Formatting, style]** The paper has a high number of formatting, stylistic, and wrong element reference issues.
> - We apologize for the inconvenience. We've conducted another round of proofreading to improve the writing quality across the entire manuscript, incorporating your comments. As ICLR allows updating the pdf, we've updated the final PDF for better clarity. Please review the updated version.
>
>
> **[W3. Human study]** The human study provides more insight around participant perception of copyright infringement, than actual determination of violation determination.
> - We didn’t train participants to make legal judgments about copyright infringement because **it's too complex for non-experts**.
> Instead, participants identified identical or similar images, as shown in Figure 15. While actual violation determinations may differ, our goal is to highlight potential copyright infringement risks that can be easily identified by non-professionals (Lines 491-494).
>
> **[W4. Release date or version for T2I models]** Commercial T2I models are updated continuously. Please include the release date or version of each T2I model.
> - Original experiment was tested on a ChatGPT (ChatGPT-4 version) (Line 754). We employ gpt-4-0125-preview, gpt-3.5-turbo-0125, and dall-e-3 version for generating the prompt. We add this in the revised manuscript.
> - All rebuttal experiments were tested with a version of ChatGPT (ChatGPT-4o version).
>
> **[W5. Dataset release]** One of the paper's contributions is a dataset, though the dataset was not included as an anonymized link for review.
> - Thanks to your comment, we additionally **provide the dataset in the supplementary file**.
>
> **[Q1: "Identical violations" was mentioned twice in the paper but not defined.]** In lines 359-360, authors state that "Upon examining the images classified as identical violations, it was found that over 50% were deemed to be cases of copyright infringement in product and logo." If identical violations refer to generating nearly identical images, the number of over 50% being deemed as infringement seems low.
> - We believe that a 50% rate of identical violations is alarmingly high from our perspective.
> - **This issue's severity depends on the perceived importance of copyright infringement.** Since copyright is legally protected, we believe T2I services—especially commercial ones—should ensure 100% compliance. However, current systems violate copyright in over 80% of cases, with 50% involving severe infringements. We find this deeply problematic.
>
>
> **[Q2: Lines 371-372: "In Figure 4, 42.19% of the generated images correctly answer more than seven questions."]** What are the seven questions?
> - 10 questions are generated with VLM based on given target images (Line 968). The LLM answers more than seven questions correctly for 42.19% of the generated images.
> - Seven questions vary depending on the target image.

---

> > ### Author Response · Authors · 2024-11-25
> > **Gentle reminder / [Multiple runs / Increased dataset size / Proofread / Dataset released]**
> >
> > **This is a gentle reminder regarding your initial concerns.**
> >
> > We have addressed them through additional experiments and revisions, and we kindly invite you to review our response.
> > If you have any remaining concerns or suggestions, **we would be delighted to engage in a final discussion to ensure a fair review process.**
> >
> > We greatly appreciate your time and effort and look forward to hearing from you.

---

> > > ### Comment · Reviewer_Z4wV · 2024-11-26
> > >
> > > Thank you for supplying the dataset, sharing additional experimental results, and addressing my comments.
> > >
> > > W1:
> > > - Thank you for experimenting with more iterations and more images. For these two rows, do you have baselines for simple prompt? The current baseline for 70 images for copilot is 5% block rate (Table 1 and 3). Here, the new results for block rate after attack, for 140 images for copilot has the average of 11.38%. Is the copilot baseline for 140 images much higher than 11.38%?
> > >
> > > W2, W3, W4, W5: Thank you for addressing my comments and revising the manuscript for clarity.
> > >
> > > Q1: I recommend rewording the sentence about the 50% statistic. Right now, the wording makes it sound like only some identical violations are copyright infringement (over 50% are, while the rest aren't).
> > >
> > > Q2: I see that in table 4, around 10% target images (ground truth copyrighted images) has 4 questions matched out of 10 via automatic QA evaluation. 4/10 seems low for the target images, and there's a range for the numbers of QA question match. Is there any analysis on how correlated human evaluation and QA evaluation are? Or perhaps comparison of automatic QA score for images generated by your method, and images generated by simple prompt/ablated prompts?

---

> ### Author Response · Authors · 2024-11-28
> **Additional experiment on baselines / QA evaluation and human rate correlation**
>
> W1. Thank you for experimenting with more iterations and more images. For these two rows, do you have baselines for simple prompt? The current baseline for 70 images for copilot is 5% block rate (Table 1 and 3). Here, the new results for block rate after attack, for 140 images for copilot has the average of 11.38%. Is the copilot baseline for 140 images much higher than 11.38%?
> * Yes, Copilot has updated its copyright protection, as shown in the following table. **We have also included additional baselines using simple prompts for both Copilot and ChatGPT**. With these updates, the block rates for simple prompts have improved compared to the original baselines. However, both systems remain vulnerable to our APGP prompts. The results provided represent the averaged block rates over three trials for consistency.
>
> |140 images|Product|Logo|Character|Art|Average|
> |-|-|-|-|-|-|
> |Copilot-baseline|62.86$\pm$52.78|79.05$\pm$21.44|82.86$\pm$29.69|89.34$\pm$18.47|78.53$\pm$30.60|
> |Copilot-ours|6.67$\pm$6.60|7.62$\pm$5.95|1.90$\pm$3.30|29.33$\pm$8.33|**11.38**$\pm$6.04|
> |ChatGPT-baseline|19.05$\pm$|99.05$\pm$|100.0|100.0|79.52$\pm$8.66|
> |ChatGPT-ours|5.71$\pm$2.86|15.24$\pm$5.95|26.67$\pm$22.19|24.00$\pm$12.00|**17.90**$\pm$10.75|
>
>
> Q1. Thank you for the feedback. We will clarify the sentence as follows:
> * Over 50% of the images classified as identical violations were deemed cases of nearly identical copyright infringement, while the remainder were categorized as similar infringements, particularly in product and logo categories.
>
> Q2. I see that in table 4, around 10% target images (ground truth copyrighted images) has 4 questions matched out of 10 via automatic QA evaluation. 4/10 seems low for the target images, and there's a range for the numbers of QA question match. Is there any analysis on how correlated human evaluation and QA evaluation are? Or perhaps comparison of automatic QA score for images generated by your method, and images generated by simple prompt/ablated prompts?
>
> * The range in QA matches arises from variations in question and answer quality, influenced by the capabilities of the LLM models. The **correlation between human evaluation and automatic QA evaluation is 0.422**, indicating a positive relationship. This supports the use of automatic evaluation as a cost-efficient and valid approach for preliminarily identifying copyright infringement (Line 364).
> * Additionally, the average QA score for our method is 5.63, compared to 4.81 for methods without the $S_{qa}$ score function, further demonstrating that QA evaluation effectively reflects differences in copyright infringement quality.
>
> If you have any further questions, please feel free to reach out. If our responses have addressed all your concerns, we kindly hope you will consider reflecting this in the final scores.

---

> > ### Comment · Reviewer_Z4wV · 2024-12-01
> >
> > - [W1] Thank you for the updated baseline and method results on 140 images, and clarifying that the difference in results from the current table in the paper is due to version differences for copilot. This addresses my previous concern.
> > - [Q2] It's clear that the method proposed produces higher score under the proposed QA set-up. Though, I'm still concerned that there is not an easy way to interpret the QA score (i.e., at what score cutoff, is the image highly likely to be violating copyrights?)
> >
> > Given that the results from this discussion will be incorporated into the manuscript, I am updating my score to 5.

---

> > > ### Author Response · Authors · 2024-12-01
> > > **QA score is an initial indicator of potential infringement risk**
> > >
> > > Thank you for your feedback.
> > > We're glad to have addressed your initial concerns. We also appreciate your insights on Q2 and would like to provide more details on how to interpret the QA score.
> > >
> > > Addressing copyright infringement in T2I models is extremely challenging, which explains the limited research despite its legal significance. **One major difficulty is evaluating potential infringements**, as only legal authorities can definitively determine whether content violates copyright laws. However, **involving legal judges in every experimental iteration is impractical for researchers.**
> > >
> > > The QA score is designed as an **automatic, cost-effective tool to supplement human evaluation**. While it cannot replace legal judgment, the QA score serves as an **initial indicator of potential infringement risk**. Given the positive correlation between the QA score and human evaluations, it could serve as an effective preliminary screening tool to flag content that may require further, more in-depth analysis.
> > >
> > > Therefore, **when the QA score exceeds the baseline, it signals the need for more comprehensive analysis during red-teaming**.

---

### Meta-Review · Area_Chair_4F5K · 2024-12-26

**Metareview:**

The submission "Automatic Jailbreaking of Text-to-Image Generative AI Systems for Copyright Infringement" proposes a prompt generation pipeline aimed at the task of overcoming defenses against copyright infringement in place by commercial image generation systems. The core output of this pipeline is a detailed description of the target, copyrighted, image. These descriptions most often break systems designed to catch more explicit call-outs to the copyright image or concept.

Aside from a number of concerns regarding the writing and style of the submission, reviewers also point out the small-scale nature of the study (using 70/140 targets), and question, among other, smaller concerns, the validity of using non-export human reviewers in the study. There is an underlying question here whether the generated prompts are even a concern for copyright law. Arguably the API providers did already provide some level of due dilligence by blocking egregious requests. Whether the generation of, e.g. Mickey Mouse, after careful description of all details of the character is a liability of the API provider, or the user, is not so clear.

Given that the proposed pipeline is not a large methodological contribution, these questions are the kind of analysis I would have wished this kind of paper to contain. Due to these questions, and due to only muted support from reviewers, I am not recommending acceptance of this work for now.

**Additional Comments On Reviewer Discussion:**

After extensive discussion with the authors, the reviewers raised some of their scores, such as Z4wV, who raised their conclusion to "marginally below the acceptance threshold", after the authors include 70 new data points, and reviewer Lo6a who raised their score to "marginally above the acceptance threshold" after discussing presentation issues, ablations and dataset size. Reviewer GDMQ remains below the acceptance threshold with concerns regarding ablation studies and the strength of the unlearning experiments remaining after discussion.

---

### Decision · Program_Chairs · 2025-01-22

Reject